# Epitranscriptomics as a New Layer of Regulation of Gene Expression in Skeletal Muscle: Known Functions and Future Perspectives

**DOI:** 10.3390/ijms242015161

**Published:** 2023-10-13

**Authors:** Carol Imbriano, Viviana Moresi, Silvia Belluti, Alessandra Renzini, Giorgia Cavioli, Eleonora Maretti, Susanna Molinari

**Affiliations:** 1Department of Life Sciences, University of Modena and Reggio Emilia, 41125 Modena, Italy; silvia.belluti@unimore.it (S.B.); eleonora.maretti@unimore.it (E.M.); 2Institute of Nanotechnology, National Research Council (CNR-NANOTEC), University of Rome “La Sapienza”, 00181 Rome, Italy; viviana.moresi@cnr.it; 3Unit of Histology and Medical Embryology, Department of Human Anatomy, Histology, Forensic Medicine and Orthopedics, University of Rome “La Sapienza”, 00161 Rome, Italy; alessandra.renzini@uniroma1.it (A.R.); giorgia.cavioli@uniroma1.it (G.C.)

**Keywords:** epitranscriptomics, RNA modifications, m6A, skeletal muscle, gene expression

## Abstract

Epitranscriptomics refers to post-transcriptional regulation of gene expression via RNA modifications and editing that affect RNA functions. Many kinds of modifications of mRNA have been described, among which are N6-methyladenosine (m6A), N1-methyladenosine (m1A), 7-methylguanosine (m7G), pseudouridine (*Ψ*), and 5-methylcytidine (m5C). They alter mRNA structure and consequently stability, localization and translation efficiency. Perturbation of the epitranscriptome is associated with human diseases, thus opening the opportunity for potential manipulations as a therapeutic approach. In this review, we aim to provide an overview of the functional roles of epitranscriptomic marks in the skeletal muscle system, in particular in embryonic myogenesis, muscle cell differentiation and muscle homeostasis processes. Further, we explored high-throughput epitranscriptome sequencing data to identify RNA chemical modifications in muscle-specific genes and we discuss the possible functional role and the potential therapeutic applications.

## 1. Epitranscriptomics at a Glance

Epitranscriptomics is a recently born field that studies the multitude of biochemical post-transcriptional RNA modifications and editing which give rise to functionally relevant changes in the transcriptome [1]. The first discovery of RNA modification dates back to 1957, with the identification of pseudouridine (*ψ*), a nucleotide variant with respect to the four standard bases of RNA [2]. Since then, thin-layer chromatography and high-performance liquid chromatography coupled to mass spectrometry (HPLC-MS) have enabled identifying more than 170 chemical modifications for both coding and non-coding RNAs in various species, humans included (listed at https://iimcb.genesilico.pl/modomics/ (accessed on 15 June 2023)) [3]. The variety of chemical modifications observed in RNAs is much wider than the 17 covalent modifications detected in genomic DNA [4], conferring a high grade of versatility to the RNA molecules that need to perform a large number of regulatory and catalytic functions through the assumption of specific folds. RNA structures are generated by the three-dimensional organization of small structural motifs determined by base pairing between complementary sequences. To expand the structural flexibility of RNA, folding is based on different types of base pairings, which are established through hydrogen bonding interactions between different edges of the base molecules: the Watson–Crick edge, the Hoogsteen edge and the sugar edge. Chemical modifications of RNA are installed on all three edges, further increasing the functional diversification of RNA species. Furthermore, non-canonical base pairing generates functional variation in the exposed surfaces for interacting protein ligands [5]. Thus, chemical modifications have an impact on RNAs’ base pairing potential and ultimately on their folding and protein–RNA interactions.

In most cases, the functional importance of RNA modifications has been demonstrated and correlated with the function of the modified RNA species. For example, base modifications were first identified in abundant RNA species such as transfer RNAs (tRNAs) and ribosomal RNAs (rRNAs) in the 1960s, and their possible role in RNA processing and size reduction has been inferred [6,7,8]. Thereafter, it became evident that chemical modifications occur in all types of RNAs, including messenger RNA (mRNAs), small nuclear RNAs (snRNAs), small nucleolar RNAs (snoRNAs), long non-coding RNAs, microRNAs, and circular RNAs.

At first, it was assumed that RNA modifications were constitutive and always present in the same positions, fine-tuning the chemo-physical properties, the structure, and the catalytic function of RNAs. However, in the last decade, it has been shown that many RNA decorations are reversible, dynamic, and sensitive to external stimuli; in some cases, it has also been suggested that a possible crosstalk between diverse RNA modifications exists [1,9]. Epitranscriptomics studies revealed that post-transcriptional RNA modifications regulate all facets of RNA metabolism, comprising RNA processing, export, translation, and stability, thus representing an additional layer of gene regulation. Recent evidence also implies a role for RNA modifications in DNA double-strand break repair [10].

RNA modifications are the result of the activities of highly conserved enzymes involved in the establishment of the modifications (the writers or effectors) and enzymes that reverse the modifications (the erasers). RNA-binding proteins (the readers, also known as binders) recognize and bind to the modified RNAs, thus modulating their fate and function. The RNA Modification Enzyme (RNAME) database (https://chenweilab.cn/rname/ (accessed on 18 June 2023)) has been developed to provide a comprehensive resource for the enzymatic machineries responsible for RNA modifications [11]. These three classes of proteins are distributed throughout the nuclei, cytoplasm and mitochondria and control several biological processes; hence, their disruption is linked to a variety of diseases [12]. The isolation of antibodies directed against specific RNA modifications, combined with next-generation sequencing, has strongly influenced the field of epitrascriptomics, especially to also probe RNA modifications in the sequence context in low-abundant RNA species such as mRNA. It is also possible to identify RNA modification at a single-nucleotide resolution by using labeling strategies that take advantage of the unique chemistry of modified bases. Due to space limitations, we do not discuss the techniques utilized to map nucleosides modifications, which have been extensively reviewed elsewhere [13,14,15]. A further great impetus to this field of study came from the discovery that some chemical modifications are implicated in critical biological processes and the RNA modification machinery is often dysregulated in human diseases, including cancers, suggesting that the modulation of the enzymatic activities involved has a potential therapeutic application in oncological diseases [16].

Approximately ten modifications have been described in mammalian mRNA, among which the most represented and characterized is N⁶-methyladenosine (m6A), which regulates most aspects of mRNA metabolism and is implicated in several biological processes. The latest advances in epitranscriptomics have revealed the importance of RNA modifications other than m6A in regulating many features of RNA processing.

This review focuses on epitranscriptomics in skeletal muscle, where, to date, only m6A modification has been described. Of note, an analysis performed on data collected in the RMBase v3.0 database [17], a platform for exploring RNA modification sites derived from high-throughput epitranscriptome sequencing data, underscored the presence of six additional RNA modifications in the transcripts encoding proteins that are implicated in skeletal muscle homeostasis [18,19]. In the first part of this review, we will provide an overview of the current knowledge about m6A and the six supplementary RNA modifications reported in the RMBase v3.0 database. In the second part, we aim to provide an overview of the literature regarding the functional meaning of m6A in skeletal myogenesis. Lastly, we will speculatively discuss the possible functional role of RNA chemical modifications in muscle-specific gene expression and the consequent therapeutic applications.

### 1.1. N⁶-Methyladenosine (m6A)

(Figure 1A, Table 1). m6A is the most prevalent modification in the 5′ CAP of mRNA, where it has a key role in its stabilization; its deposition depends on the activity of the 5′-CAP-specific adenosine N6 methyl transferase (CAPAM) [20,21]. Adenosine methylation has also been observed in internal sites of mRNAs in the 1970s. Since then, it has been found to be the most prevalent chemical modification in the transcripts of many species, including mammals (0.2–0.6% of total adenosines) [22,23,24]. In addition to mRNA, m6A is observed in all RNA types, including tRNA, rRNA and non-coding RNA (ncRNA). At the molecular level, m6A results in the regulation of stability, conformation or interaction of RNA with binding proteins. m6A does not impede canonical Watson–Crick A:U base pairing but can disrupt the formation of trans-sugar-Hoogsteen G:A base pairs, which are abundant within the core of kink turns (k-turns), the RNA structural motifs that function as protein binding sites [25,26]. m6A is a selective modification, which occurs inside the consensus sequence DRACH (D = A, G, or U; R = G or A; H = A, C or U) [27,28]. High-throughput sequencing revealed that m6A is statistically more represented near the distal end of mRNAs, around the stop codon, and in the 3′ untranslated region (UTR) [29,30]. Moreover, m6A has been reported within the coding region and in the 5′UTR of mRNA, where it can enhance CAP-independent translation initiation [31]. Several transcription-based and chromatin-based hypotheses have been made about the mechanisms underlying the selection of specific transcripts by the methyltransferase complex, ranging from the intervention of transcription factors that mediate the interaction between the target RNA and the catalytic complex, to the number of CpG islands in the upstream promoter or the type of histone modifications [32].

Adenosine methylation is transcript specific and highly heterogeneous from cell to cell. Its abundance is mediated by external stimuli and depends on the balance between the activities of two classes of enzymes: methylases and demethylases. A great stimulus in the study of the functional role of m6A modification came from the identification of the nuclear multi-subunit methyltransferase complex, which includes the methyltransferase-like 3 (METTL3)–methyltransferase-like 14 (METTL14) heterodimer, with METTL3 being the catalytic subunit and METTL14 being its allosteric activator, the Wilms Tumor 1-Associating Protein (WTAP), the Vir-Like M6A Methyltransferase Associated (VIRMA) protein, also indicated as KIAA1429 and the Zinc Finger Protein 217 (ZFP217) [33]. The METTL3–METTL14 methyltransferase complex accounts for the deposition of m6A in newly synthesized RNA polymerase II (Pol II)-dependent transcripts. More recently, an additional m6A mRNA methyltransferase, METTL16, has been isolated, which regulates the cellular levels of S-adenosyl-methionine (SAM), a substrate of methylases that functions as a methyl donor, and targets the U6 small nuclear RNA (U6 snRNA), which regulates mRNA splicing [34]. The reversibility of m6A is ensured by the activity of the demethylases α-ketoglutarate-dependent dioxygenase alkB homolog 5 (ALKBH5) and fat mass- and obesity-associated protein (FTO) [9,35]. Methylated transcripts are subsequently regulated by m6A readers that modulate various aspects of their processing. Accordingly, the main mechanism underlying the regulatory effect of m6A on target RNA function is based on its ability to recruit reader proteins. The major class of m6A readers is represented by the family of YT521-B homology (YHT) proteins that contain a ~150 amino acid-YTH domain that allows RNA recognition in an m6A-dependent manner through a “tryptophan cage”, in which two or three tryptophan residues wrap the methyl group. YTH proteins interact with m6A-containing RNAs both in the nucleus (YTHDC1 and YTHDC2) and in the cytoplasm (YTHDF1, YTHDF2, and YTHDF3). YTH proteins may also contain a low-complexity region, which triggers the phase separation of YTH proteins, especially when bound to m6A mRNA, thus targeting methylated transcripts to non-membranous P bodies, stress granules, and other RNA–protein complexes where they are processed. The effects of the YTH binders are disparate and dependent on the specific reader that binds the modification. Additional m6A binders have been isolated: proline-rich coiled-coil 2 A (PRRC2A) binds and stabilizes oligodendrocyte transcription factor 2 (*Olig2*) mRNA, which controls oligodendrocyte specification; this m6A-dependent interaction involves a consensus GGACU motif in the coding sequence of *Olig2* [36]. Other readers can bind to unfolded RNA induced by the presence of m6A. While the methyl group at the N6 position of adenosine does not alter Watson–Crick A–U base pairing, the steric hindrance of the methyl group on the Watson–Crick edge stabilizes unpaired bases, thus facilitating the exposure of binding sites for, e.g., insulin-like growth factor 2 binding proteins (IGF2BPs) and fragile X mental retardation protein (FMRP) [37,38,39]. IGF2BP1, 2, and 3 are single-stranded RNA-binding proteins that contain six RNA-binding domains: two RNA recognition motif (RRM) domains and four K homology (KH) domains. Upon binding to m6A-modified transcripts, IGF2BPs promote their stability [37]. Accordingly, the first function identified for m6A was its ability to regulate mRNA stability, both negatively through interaction with YTHDF2 and positively by the mediation of IGFBPs [26]. Another m6A reader is represented by the heterogeneous nuclear ribonucleoprotein C (HNRNPC), an abundant nuclear RNA-binding protein responsible for pre-mRNA processing. This interaction explains the ability of m6A to modulate alternative splicing of pre-mRNAs [40]. Similarly, HNRNPA2B1 is a m6A reader involved in primary microRNA processing and alternative splicing [41]. Regulation of alternative splicing by m6A is also linked to its interaction with the YTHDC1 reader that can recruit the splicing regulators and arginine-rich splicing factors 3 (SRSF3) and 10 (SRSF10) [40,42]. In addition, m6A can promote mRNA nuclear export and regulate translation in different ways depending on the position of the chemical modification [26,39,43,44].

The analysis of the distribution of m6A shows that this modification is enriched in the transcripts of genes that regulate development and cell fate specification, while it is rare in the transcripts of housekeeping genes [29,45]. Accordingly, m6A is implicated in a wide range of key biological functions and aberrant m6A distribution, together with mutations in genes encoding proteins involved in RNA modification, are often associated with severe diseases, including cancer, metabolic, neurological and cardiac disorders [12,46,47,48,49,50].

### 1.2. N1-Methyladenosine (m1A)

(Figure 1B, Table 1). m1A is an m6A isomer that results from a methylation occurring on N1 of adenosine. This chemical modification confers a positive electrostatic charge under physiological conditions and impairs the canonical Watson–Crick base pairing through electrostatic and steric effects. This is an ancient modification that is conserved across species and is present in all kinds of RNAs, including cellular and mitochondrial mRNAs [51,52,53]. The presence of m1A has been well characterized in tRNA and rRNA, where it has a role in maintaining the tertiary structure [54,55]; in mRNAs m1A is 10-fold less represented than m6A (0.01–0.05% of total adenosines). Several transcriptome-wide mapping experiments have been undertaken to define the m1A methylome [51,52,56,57,58]. The first studies were based on next-generation sequencing of antibody-mediated immunoprecipitated RNAs. These works identified many m1A sites in mRNA, enriched around both canonical and alternative translational initiation sites and in GC-rich highly structured regions around the start codons, and these were linked to enhanced translation [51,52]. Other studies instead described m1A in a low number of cytosolic mRNAs, mostly within the codon region and 3′UTR, as having a repressive role on translation when detected in coding sequences (CDS) [56,58]. The divergent results have been interpreted because of the low specificity of the antibody directed against m1A. The altered physical-chemical properties of m1A can regulate protein translation in different ways that depend on the location of m1A. When m1A is allocated in the 5′UTR, translation is enhanced possibly through the destabilization of the secondary structure in the 5′ UTR, while m1A in the CDS interferes with translation [59]. Similar to m6A, m1A is a dynamic modification that is regulated by external stimuli: for example, it is promoted by oxidative damage and starvation [51,52]. m1A might have a protective role on RNA that accumulates in stress granules upon heat shock [60]. Deposition of m1A in tRNA is ensured by the tRNA methyltransferases TRMT6–TRMT61A complex, which also catalyzes methylation of adenosine in mRNAs that contain a tRNA T-loop structure [57]. A methyltransferase complex including TRMT61B and TRMT10C is responsible for the deposition of m1A marks in mitochondrial RNA [61]. Ablation of m1A is catalyzed by the alkB family of demethylases, and specific demethylation of m1A in mRNA is catalyzed by ALKBH3 [52]. The cellular readers of m1A are still under study, but in vitro data indicate that m1A can be recognized and bound by YTH proteins (YTHDC1, YTHDF1, YTHDF2, YTHDF3) [62], thus suggesting a crosstalk between m6A and m1A RNA modifications.

### 1.3. 5-Methylcytidine (m5C)

(Figure 1C, Table 1). In m5C modifications, a methyl group is attached to the fifth carbon of the cytosine ring, a chemical imprint that was first detected on DNAs and later on RNAs in the 1970s [63]. The development of bisulfite-based techniques to identify m5C mark in RNAs has enabled the detection of its deposition in all types of RNA molecules. In mRNAs, it is 3–10-fold rarer than m6A (0.03–0.1% of cytosine), and it is enriched in the 5′ or 3′ UTRs or next to translational start sites [64,65,66]. m5C deposition does not alter the base pairing edges, but it significantly changes the physicochemical properties of the original nucleobase, modulating its ability to interact with proteins [67]. m5C deposition in RNAs is catalyzed by the tRNA aspartic acid methyltransferase 1 (TRDMT1), also called S-adenosyl-methionine-dependent methyltransferase 2 (DNMT2), and the nucleolar protein 1 (NOL1/NOP2)/Sun domain (NSUN) methyltransferase protein family, which includes seven proteins in humans (NSUN1 to NSUN7), with NSUN2 and NSUN6 being the responsible members for m5C deposition in mRNAs [53,68,69,70]. To date, the erasers of mRNA m5C marks are still under debate. While a demethylation pathway for m5C modifications of DNA has been described that involves the ten–eleven translocation (TET) demethylases, the involvement of TET demethylases as m5C erasers is still unclear [71,72].

Interestingly, a positive reciprocal interaction exists between m5C and m6A [73]. m5C can increase mRNA stability and promote mRNA export from the nucleus to the cytoplasm; furthermore, m5C modulates protein translation, both positively and negatively [74,75,76]. These activities are ensured by the interaction with the readers Aly/REF export factor (ALYREF) and Y-box binding protein 1 (YBX1). Further, RAD52 is a reader of m5C marks, which is involved in the DNA damage pathway by recognizing hybrid strands containing m5C-marked RNAs and DNAs [77]. Aberrant m5C deposition on RNA has been linked to several diseases [78] and 5-hydroxymethylcytosine (hm5C) has been suggested to play an important role in the regulation of embryonic stem cell differentiation [79,80,81].

### 1.4. 7-Methylguanosine (m7G)

(Figure 1D, Table 1). m7G was first identified as a component of the CAP structure of Pol II transcripts. The m7G CAP modification was first identified on eukaryotic mRNAs, and subsequent studies have shown that this is an evolutionarily conserved modification in all organisms. It is installed co-transcriptionally on nearly all RNA polymerase II target genes and it represents a critical feature required for stability, splicing, and efficient translation of mRNAs [82,83,84,85]. m7G deposition in the CAP of mRNA is catalyzed by the mRNA CAP methyltransferase RNMT in complex with the RNMT-activating mini-protein (RAM) [86,87]. m7G is also found internally in the variable loop of tRNAs and in eukaryotic 18S rRNAs, where it modulates RNA processing and function. Aberrant m7G has been linked to human diseases, such as microcephalic primordial dwarfism [88]. Installation of m7G in tRNAs and rRNAs is catalyzed by a complex of METTL1 with WD repeat domain 4 (WDR4) and a complex of Williams–Beuren syndrome chromosome region 22 (WBSCR22) protein, also known as BUD23, with tRNA methyltransferase activator subunit 11–2 (TRMT112), respectively [55,89,90,91]. In addition to its ubiquitous presence in the CAP, m7G is also found internally in mRNAs as it has been first evidenced by differential enzymatic digestion combined with HPLC-MS analysis [92]. Subsequent works enabled the mapping of thousands of m7G marks in mammalian mRNAs enriched both in the 3′ UTR and in the 5′ UTR, by m7G-methylated RNA antibody-based immunoprecipitation sequencing (MeRIP-seq) and chemical-based methods [93,94]. These studies revealed that internal mRNA m7G promotes mRNA translation and is dynamically regulated upon heat shock and oxidative stress [95]. m7G has also been detected in human mature miRNAs and miRNA precursors, where it is important to produce mature miRNAs and maintain high levels of mature let-7e miRNAs [96]. Writers, erasers, and readers of internal mRNA m7G are still unidentified, even if a subset of m7G marks is deposed by the METTL1-WDR4 complex [93]. Mechanistically, this chemical modification takes place in the Hoogsteen edge of guanosine and, similarly to m1A, introduces a positive charge that could potentially modulate protein–RNA interactions and reorganize local secondary structures in RNAs, and consequently their biological functions. For example, it has been suggested that m7G could inhibit the formation of G-quadruplex structures, which are four-stranded structures based on the Hoogsteen base pairing of guanosines that play key roles in the control of gene expression [97]. Prevention of G-quadruplex is important to guarantee let-7e pri-miRNA processing [96].

### 1.5. Pseudouridine (ψ)

(Figure 1E, Table 1). Pseudourydilation is an isomerization reaction whereby uridine undergoes a 180° rotation, resulting in the formation of a glycosidic bond between the C5 of uridine and C1′ of the ribose sugar, which substitutes the N1-C1′ glycosidic bond of uridine. This modification is conserved in various species; it is the most represented post-transcriptional modification in all types of RNAs, so much so that it earned the designation of “fifth ribonucleotide” [2]. Installation of *ψ* increases the hydrogen bonding potential of the ribonucleotide that exhibits an extra hydrogen donor (N1 imino proton) and does not change the Watson–Crick base pairing property. Consequently, the presence of *ψ* influences the physical-chemical properties of the modified RNA. In tRNAs, *ψ* is important for maintaining its structure, while in rRNAs, it influences the interaction with tRNAs and mRNAs, thus regulating translation fidelity [98,99]. Spliceosomal snRNAs are pseudouridylated, suggesting a role in the assembly of the spliceosomal machinery [100]. This modification has also been observed in mRNAs (0.2–0.6% of the total uridines in mRNA), both in the CDS and in the 3′UTR. The number of *ψ*s is dynamically regulated by external stimuli [30,101]. In mRNAs, *ψ* promotes read-through at *ψ*-containing stop codons through non-canonical base pairing [102]. Another function is related to the regulation of RNA splicing: insertion of *ψ* in the pre-mRNAs may happen in intronic sequences that are critical for the interaction with the spliceosomal machinery; furthermore, *ψ* influences alternative splicing when it decorates retained introns and cassette exons, or RNA-binding protein (RBP) binding sites critical for splicing [103,104]. The *ψ* writers belong to a large family of pseudouridine synthases (PUS), which are generally RNA-independent enzymes that modify uridine by recognizing a consensus sequence and/or secondary structural elements of the target RNA (reviewed in [53]). In contrast, dyskerin pseudouridine synthase 1 (DKC1) is an RNA-dependent enzyme that is recruited to the target RNA by a small nucleolar RNA guide. In the PUS family of enzymes, PUS1, PUS7 and RNA pseudouridine synthase 4 (RPUSD4) can depose *ψ* cotranscriptionally [104]. There are currently no known erasers or readers for this RNA modification. Aberrant pseudouridylation has been associated with human cancers and other diseases such as mitochondrial myopathy, lactic acidosis, and sideroblastic anemia (MLASA), an autosomal recessive disease in humans characterized by disorders of the oxidative phosphorylation and iron metabolism in skeletal muscle and bone marrow [105].

### 1.6. 2′O-Methylation (Nm or 2′-O-Me)

(Figure 1F, Table 1). This is a conserved RNA modification that is characterized by the methylation of the 2′ hydroxyl (–OH) group of the ribose. O-Me is not limited to a specific base; therefore, it is indicated as Nm, where N stands for any nucleoside (Am, Um, Cm, or Gm). Nm sites have been detected in all types of RNAs: they increase the hydrophobicity of target RNAs, protect them from nuclease attacks, regulate RNA folding, and impact the ability of modified RNA to interact with proteins or other RNAs (reviewed in [106]. rRNA contains several Nm sites located in key functional sites of the ribosome [107,108,109,110]. 2′O methylation is also located at fixed positions in tRNAs, and it is linked to tRNA stability and translation efficiency [111,112]. 2′-O methylation in mRNA is observed in the CAP region, on the first and second transcribed nucleotides [113]. In addition, there are also internal Nm sites that have been identified with high-throughput mapping: they are widespread, without preference for any region of the mRNA, representing roughly 0.01% of their unmodified counterparts [114,115]. Nm sites in mRNA regulate mRNA stability, translation, and its ability to interact with other RNAs; in addition, 2′-O methylation of mRNAs has been shown to regulate gene expression in vivo [116]. There are two classes of Nm writers: independent methyltransferases and small nucleolar ribonucleoprotein (snoRNP) complexes which include the C/D-box small nucleolar RNAs (C/D snoRNAs); these complexes are indicated C/D snoRNP. Small RNAs mark the target nucleotide sequence to be methylated by the enzyme fibrillarin (FBL) through sequence complementarity. An unbiased motif search of Nm sites has revealed the presence of a consensus sequence next to the Nm which contains an AGAUC sequence followed by a 5 nt-long AG-rich stretch, suggesting the existence of a specific methyltransferase that might function independently from snoRNPs [114]. Independent enzymes CAP methyltransferase 1 (CMTR1) and CMTR2 catalyze Nm modifications occur in the CAP structure [106,117,118].

### 1.7. Nucleoside Editing

(Figure 1G, Table 1). The first description of RNA editing in mammals came after the discovery of cytosine-to-uridine (C-to-U) conversion in the mRNA encoding apolipoprotein-B100 that causes a premature stop codon, leading to the production of the shorter apolipoprotein-B48 isoform in vertebrate intestine [119]. C-to-U mRNA editing is catalyzed by cytidine deaminases that belong to the activation-induced cytidine deaminase/apolipoprotein B editing complex (AID/APOBEC) family [120]. In the same year, adenosine-to-inosine (A-to-I) RNA editing was described in the oocytes of *Xenopus Leavis* and shown to be crucial in the unwinding of double-stranded RNAs (dsRNAs) [121]. Subsequent studies have demonstrated that A-to-I is the most common type of RNA editing. This RNA modification is a highly conserved process that underlies multiple cellular processes [122,123,124,125]. A-to-I editing is a hydrolytic deamination catalyzed by adenosine deaminases acting on dsRNAs (ADARs). In mammals, there are three *ADAR* genes: ADAR1 and 2 are catalytically active, while ADAR3 is inactive and may function as a dominant negative [126]. In *Drosophila*, mice, and humans, A-to-I editing events are strongly enriched in the brain [127,128,129,130]. Millions of A-to-I editing events have been identified in humans, and they are enriched in genes involved in neurological disorders and cancer [131]. An atlas of A-to-I editing is available (https://omictools.com/the-rna-editing-atlas-tool (accessed on 30 June 2023)). In human mRNAs, A-to-I editing is detected principally in introns and UTRs; the preferred target sequences are intronic retrotransposon elements such as Alu repeats, but it is also observed in coding regions [132,133,134,135]. It has been reported that m6A negatively regulates A-to-I RNA editing [136]. Inosine has distinct base-pairing abilities in comparison to adenosine; it can base pair with any natural bases, but it preferentially pairs with cytosine rather than with uridine, leading to variations in the secondary structure of target RNAs and modifications of the encoded information. Inosines are read as guanosines by the ribosome; therefore, A-to-I editing can change the amino acid sequence of proteins [126]. MicroRNA A-to-I editing can have an impact on RNA splicing, when it takes place in sites such as the RNA splicing sites [137], or it can change the target specificity of miRNAs, thus impacting RNA degradation [126,138].

## 2. Epitranscriptomics in Skeletal Muscle

### 2.1. N⁶-Methyladenosine Modification in Embryonic Myogenesis

The importance of m6A in modulating embryogenesis comes from several studies in which the deletion of the m6A methyltransferase *Mettl3* is embryonic lethal in mice [139], due to a compromised transition from self-renewal to differentiation state in embryonic stem cells and to the targeting of several transcripts encoding pluripotency transcription factors [139,140,141]. In addition to METTL3, another N6-methyltransferase, METTL16, is important in embryonic development [142]. Indeed, *Mettl16* knockout in mice causes developmental arrest around the time of implantation by influencing the mRNA levels of the SAM synthetase methionine adenosyl transferase synthetase 2A MAT2a [142]. Similarly, embryonic neural stem cells lacking METTL14 display markedly decreased proliferation and premature differentiation, suggesting that m6A modification affects embryonic neural stem cell self-renewal [143]. In addition to methyltransferases, ZFP217, a factor that modulates m6A deposition, is crucial for embryonic stem cell fate, since its knockdown results in compromised cell growth and lineage differentiation by regulating the transcripts of pluripotency-associated factors [144]. These findings confirm that m6A modification is important for embryonic stem cell self-renewal maintenance and mouse development.

Numerous studies point to the m6A modification as an important epigenetic mechanism regulating muscle development during embryogenesis. Highly dynamic changes in RNA m6A modification have been profiled across different stages of skeletal muscle development [145,146], finding m6A modification on transcripts of important genes for skeletal muscle development. For instance, MeRIP-seq analysis identified several differentially methylated genes enriched in pathways related to porcine skeletal muscle development [147]. Similarly, the analysis of m6A distribution in Dingan goose [148], goat [149], and duck [150] embryonic skeletal muscle revealed differentially regulated m6A peaks in important developmental pathways, including the mitogen-activated protein kinase (MAPK) and Wnt signaling, or in muscle-related genes during key embryonic stages. Moreover, RNA immunoprecipitation (RNA IP) assay revealed that the m6A methylation reader IGF2BP1 targets many embryonic myogenic genes in porcine skeletal muscle, including the myogenic marker myogenin and the terminal differentiation gene myosin heavy chain 2 [147]. Coherently, the deletion of the m6A demethylase *Fto* gene in mice during the pregnancy period results in fewer and smaller myofibers, if compared to controls, further indicating its involvement in the skeletal muscle development of the offspring [151].

Taken together, these studies clearly suggest that m6A modification affects skeletal muscle development in numerous species. The biggest limitation of these findings is that they originate from methods based on RNAs extracted from a bulk of different cells, thus losing spatial localization information. Analytical techniques for m6A RNA with single-cell resolution and spatial information will be more informative. Recently, a m6A-specific in situ hybridization mediated proximity ligation assay (m6AISH-PLA) has been developed, which allows to visualize cellular m6A RNA at single-molecule resolution and could be used to investigate cell-to-cell variation and spatial pattern [152]. By applying this technology to histological sections, it would be possible to follow the spatial-temporal dynamics of m6A modification during myogenesis.

### 2.2. N⁶-Methyladenosine Modification in Myoblast Differentiation

In vitro studies clarified that RNA m6A methylation finely regulates the transition from myoblast proliferation to differentiation (Figure 2) [145,153,154,155]. Moreover, co-localization and interaction between the transcriptional (m5C) and post-transcriptional (m6A) modifications have been described during the differentiation process of the murine myoblast cell line C2C12 [156], revealing a cooperative regulation of m5C and m6A modifications in spatiotemporal gene expression during myogenesis.

The first publication on an epitranscriptomic modifier enzyme in myoblast differentiation goes back to 2017, when Wang and coworkers started addressing the role of the m6A demethylase FTO in C2C12 and primary myoblast differentiation [151]. FTO expression raises during myoblast differentiation, paralleling m6A demethylation. FTO silencing in primary myoblasts suppresses myogenic differentiation by affecting mitochondrial biogenesis through the mechanistic Target of Rapamycin (mTOR)-Peroxisome Proliferative-Activated Receptor, Gamma, Coactivator 1, Alpha (PGC-1alpha) axis, while FTO overexpression does not affect myotube formation [151]. A limitation of this study is the analysis of primary myoblasts in which FTO was silenced by siRNA technology, instead of knocking-out FTO in vitro in primary myoblasts isolated from the inducible skeletal muscle-specific FTO mice. Moreover, it is still unclear why FTO overexpression does not affect myoblast differentiation. Similar results were reported by other studies: indeed, FTO deficiency has been associated with a reduction in fat and lean mass in mice [157], while FTO overexpression leads to obesity without affecting skeletal muscle mass [158]. FTO knockdown in primary goat myoblasts increases cyclin D1 (*Ccnd1*) and growth arrest and DNA damage inducible beta (*Gadd45b*) mRNA m6A modification, thereby decreasing their stability and leading to impaired myoblast proliferation and myogenic differentiation [149].

Contrary to FTO, the expression levels of the methyltransferases METTL3, METTL14, and WTAP decrease during myoblast differentiation in vitro and in vivo [159]. Another pioneering study in 2017 clarified the pivotal role of the m6A methyltransferase METTL3 in maintaining the mRNA levels of a master myogenic regulator of skeletal muscle, i.e., myogenic differentiation 1 (*MyoD*), and therefore the myogenic potential throughout the cell cycle in proliferating myoblasts [160]. Indeed, METTL3 knockdown inhibits myotube formation, similarly to *MyoD* deletion [160]. Numerous recent studies focused on the role of the m6A writer METTL3 in muscle differentiation. The silencing of *Mettl3* in proliferating myoblasts induces premature C2C12 differentiation in vitro and reduces the capacity of serial transplantation in vivo [154]. Consistently, deletion of *Mettl3* specifically in muscle stem cells inhibits their proliferation and skeletal muscle regeneration upon injury, by regulating the expression of several genes involved in the neurogenic locus notch homolog protein 1 (NOTCH1) signaling pathway [161] required for muscle stem cell self-renewal, differentiation, and muscle regeneration [162]. Conversely, *Mettl3* overexpression in a muscle stem cell-specific *Mettl3* conditional knock-in mouse increases muscle stem cell proliferation and muscle regeneration in vivo following injury [161].

A faster muscle stem cell differentiation due to an altered transition from proliferation to differentiation state upon METTL3 or METTL14 knockdown was also confirmed by Kudou K. and colleagues in 2017 [159], while overexpression of either METTL3 or METTL14 inhibits myotube formation. The m6A writers METTL3 and METTL14, together with the m6A reader YTHDF1, finely regulate the mRNA of MAP kinase-interacting serine/threonine kinase 2 (Mknk2), a critical regulator of extracellular signal-regulated kinase (ERK)/MAPK signaling, thus controlling myoblast differentiation [159]. Similarly, silencing of the m6A reader *Igf2bp1* promotes C2C12 proliferation, altering their transition into a differentiating state, thus inhibiting their differentiation [147].

Overall, consistent conclusions were drawn regarding the functions of RNA methyltransferases in regulating the transition from the proliferative to the differentiating state of muscle stem cells, mainly via loss-of-function studies. On the contrary, apparent contradictory data were reported in gain-of-function studies: while muscle regeneration was increased in *Mettl3* muscle stem cell-specific knock-in mice [161], myotube differentiation was inhibited in C2C12 overexpressing METTL3 [159]. However, it is difficult to arrive to final considerations on METTL3 overexpression by analyzing different experimental models and having the underlying molecular mechanisms still uncharacterized.

More recently, it has been shown that METTL3, together with the m6A reader YTHDF1, post-transcriptionally regulates the mRNA of the *Mef2c* gene, encoding a transcription factor that promotes muscle differentiation in concert with MYOD, thus affecting bovine myoblast differentiation [163]. Moreover, a positive feedback loop has been reported, since MEF2C directly binds to and activates the *Mettl3* promoter region, further regulating bovine skeletal myoblast differentiation [163]. Indeed, the expression levels of *Mettl3* and *Mef2c* positively correlate in loss and gain-of-function experiments: METTL3 knockdown decreases MEF2C expression, while METTL3 overexpression increases MEF2C protein levels. Coherently, knockdown of the demethylase FTO in myoblasts induces a significant upregulation of MEF2C protein expression, whereas FTO overexpression results in a decrease in MEF2C, further proving the direct involvement of m6A modification in the regulation of *Mef2c* expression during myoblast differentiation.

Moreover, METTL3 represses the expression of skeletal muscle-specific miRNAs during myoblast differentiation, by indirectly regulating the expression of either transcription factors or epigenetic regulators, which in turn affect miRNA expression [164]. In addition to coding mRNA, gain- and loss-of-function experiments clarified that METTL3 positively regulates the abundance of long-non-coding RNAs, thereby affecting the expression of their adjacent mRNAs, during myoblast differentiation [165].

In conclusion, an increasing number of studies in recent years pointed to the importance of m6A modification in the orchestrated regulation of muscle stem cell differentiation and myogenesis. A deeper investigation is needed to explore the key upstream and downstream factors that regulate m6A modifications, and their cooperation with other epigenetic regulators, to elucidate the specific functional mechanism of m6A modification in regulating myogenic differentiation.

### 2.3. N⁶-Methyladenosine Modification in Skeletal Muscle Homeostasis

In addition to muscle differentiation, METTL3 has been shown to control muscle mass growth and homeostasis in post-natal life by targeting the activin receptor, thereby modulating the transforming growth factor-β (TGF-β) signaling [166]. Short-term deletion of *Mettl3* in muscle fibers clarified that METTL3 is indispensable for the hypertrophic response of skeletal muscle to mechanical overload, while long-term deletion of *Mettl3* leads to a progressive decline in skeletal muscle mass [166]. Conversely, METTL3 overexpression in newborn mice with adeno-associated viruses induces a hypertrophic response, and METTL3 overexpression in adult muscles, by electroporation, induces an enhanced hypertrophic response to overload [166].

A proof of the importance of m6A modification in maintaining skeletal muscle homeostasis also comes from a very recent paper, which claims that among liver, heart, and skeletal muscle, the latter is the most susceptible to m6A decrease with aging, which positively correlates with a decreased expression of *Mettl3* [167]. *Mettl3* deficiency leads to smaller myotubes and affected senescence of myotubes. As for the mechanism, METTL3 targets and stabilizes the mRNA of nephronectin, a matrix protein involved in cell-matrix adhesion and important for myotube fusion. Coherently, knockdown of nephronectin in myoblasts phenocopies the *Mettl3* deficiency defects [167]. Similar to in aging, the m6A mRNA global levels in skeletal muscles significantly decrease upon denervation, parallel to an increase in the expression of the m6A demethylase *Alkbh5* [168]. Importantly, ALKBH5 promotes neurogenic muscle atrophy by demethylating and thus stabilizing *Hdac4* mRNA. The HDAC4 protein in turn interacts with and deacetylates forkhead box transcription factor O3 (FOXO3), resulting in the activation of FOXO3 signaling [168].

The research has moved very fast in the last three years, clarifying the role of m6A modification in muscle differentiation and homeostasis. No information is currently available regarding the other less abundant mRNA modifications in skeletal muscle, such as the m5C and hm5C modifications in tRNAs, or the pseudouridylation of coding and non-coding RNAs, which play important roles in the regulation of non-muscle stem cell fate [169,170,171]. Future studies should address their functions to better understand skeletal muscle physiology and to provide new insights for possible therapeutic approaches aimed at maintaining muscle mass.

## 3. Epitranscriptomics as a Novel Pathogenetic Mechanism and a Potential Therapeutic Approach for Muscular Disorders

Muscular disorders include a wide range of inherited or acquired diseases affecting the muscular system. Genes encoding proteins implicated in numerous processes, such as contractility, membrane integrity, gene regulation, and metabolism, are altered in muscular diseases. Over 650 genes have been associated with monogenic neuromuscular disorders [19]. Genetic information has enabled an understanding of the molecular pathogenesis of muscular disorders, such as muscular dystrophies (MDs), which have been mapped to at least 29 different genetic loci. Despite this knowledge, MDs are still undertreated, and steroid medications represent the standard treatment to slow down disease progression.

The identification of the molecular basis of RNA modifications may provide new pathogenetic mechanisms for the basis of muscular diseases and hence drive the discovery of new therapeutic targets. Since epitranscriptomics is a dynamic and reversible regulatory mechanism, the manipulation of RNA modifications represents a promising approach for the treatment of MDs and other muscular diseases. The field of epitranscriptome-targeting drugs is still in its beginning stages, but efforts are directed towards drug discovery research on RNA-epitranscriptomics and low-molecular-weight compounds that have been shown to potentially revert defects, at least in cancer [172].

The rapid progress of new technologies, such as high-throughput sequencing, has enabled the collection of multiple data. To predict potential RNA modifications in MD-related genes, we exploited the RMBase v3.0 database [17], which integrates epitranscriptome sequencing data for the investigation of post-transcriptional modifications of RNAs. Interestingly, the vast majority of RNAs encoded by genes associated with MD [19] can be methylated at different sites, according to our research, although in cellular contexts other than muscle (Table 2).

In addition to the genes listed above that are directly related to MDs because of existing genetic defects (Table 2), we investigated RNA modifications of key genes regulating myogenesis and skeletal muscle regeneration in response to injury and genetic dystrophies, such as transcription factors and epigenetic regulators (Table 3). Their central role in muscle development and maintenance makes them excellent candidates for druggable epitranscriptome therapies in muscular diseases.

Further, in this case, the majority of myogenic genes show RNA modifications, in particular m6A. This is not surprising, taking into consideration that m6A regulates gene expression through different mechanisms, as described above [165]. m6A enrichment in muscle genes is also consistent with the results from the muscle stem cell-specific *Mettl3* conditional knockout mouse model, which affects stem cells fate and muscle regeneration after injury [161].

2′-O-methylation and adenosine to inosine RNA-editing (A-to-I) represent two of the most common RNA modifications provided by RNA-guided mechanisms and, consequently, potentially exploitable for therapeutic application. For example, site-specific box C/D-directed methylation of the branchpoint adenosine could inhibit the splicing of an intron, or site-specific methylation of a central nucleotide within a sense codon may trigger premature termination of translation [173], thus inhibiting the expression of non-functional or aberrant proteins associated with muscular diseases. One 2′-O-methylation site has been identified in the transcript of enhancer of zeste 2 (*Ezh2*) (Table 2), which encodes for a subunit of the polycomb repressive complex 2 and 3 (PRC2 and PRC3) complexes with histone lysine methyltransferase (HKMT) activity, that triggers transcriptional repression and controls the expression of muscle genes and the differentiation of satellite-cell-derived myoblasts following muscle injury [174,175]. EZH2 overexpression represses muscle gene expression and differentiation; therefore, additional 2′-O-methylation could represent a strategy to interfere with its increased expression.

The inosine RNA modification that results from the hydrolytic deamination of adenosines (A-to-I) catalyzed by the adenosine deaminase ADAR represents another interesting tool to recode transcripts and alter splicing events, thus correcting disorders at the mRNA level and restoring protein function [176]. This could be relevant for diseases resulting from G-to-A genomic single point mutations. For example, one of the most common nucleotide changes at the first intronic nucleotide of the DMD gene is a G-to-A, which disrupts the splice site consensus sequence, thus producing an abnormal transcript, as reported for Duchenne and Becker MDs [177].

*Hdac4* (histone deacetylase 4), *Sirt1* (sirtuin 1), *Mef2* and *Smad4* (SMAD family member 4) transcripts show the higher number of RNA modification sites, with m6A, m5C and A-I being more represented. Both *Hdac4* and *Sirt1* genes encode for histone deacetylase enzymes, which exert a key role in the control of gene transcription and homeostasis in skeletal muscle. SIRT1 enzymatic activity is deeply correlated with the differentiation of muscle fibers, energy homeostasis and muscle cell fate signaling. In addition to SIRT1-mediated effects on the transcription of key genes, among which is *MyoD* [178], SIRT1 is intimately linked to nutrient availability in muscle cells and controls energy metabolism [179]. Moreover, as an inhibitor of nuclear factor kappa-B DNA binding subunit (NF-KB) signaling, SIRT1 activation ameliorates muscle pathology in MD [180,181].

HDAC4 is crucial in maintaining muscle integrity upon different stimuli [182]. *Hdac4* expression is stabilized by the ALKBH5 demethylates upon skeletal muscle denervation [168]. ALKBH5-mediated m6A modification of HDAC4 triggers neurogenic muscle atrophy, since HDAC4 interacts with and activates FOXO3 [168]. Although the ALKBH5-HDAC4-FOXO3 axis is of great interest for the maintenance of skeletal muscle mass, ALKBH5 cannot be considered a potential therapeutic target for the treatment of neurogenic muscle atrophy, since HDAC4 inhibition is protective upon short-term denervation [183], but it is deleterious after long-term denervation [184] or in a chronic condition of denervation, such as in Amyotrophic Lateral Sclerosis (ALS) [185,186]. On the contrary, HDAC4 functions in DMD need to be preserved, since *Hdac4* depletion is detrimental to dystrophic muscles [187]; thus, the stabilization of *Hdac4* mRNA may be a useful therapeutic approach. Similarly, another study claimed the importance of m6A modification in *Hdac4* mRNA stabilization in sepsis-induced myocardial injury. A METTL3/IGF2BP1/m6A/HDAC4 axis has been described in cardiomyocytes, where the m6A reader IGF2BP1 enhances *Hdac4* mRNA stability and thus regulates the inflammatory damage of cardiomyocytes induced by lipopolysaccharide [188]. The identification of RNA modifications in genes encoding for chromatin modifiers, such as EZH2, SIRT1 and HDAC4, provides a proof of a crosstalk between epitranscriptional and epigenetic mechanisms in muscle physiology and pathology.

Among the transcription factors that regulate muscle gene expression, a key role is played by the MEF2 family of proteins which work in concert with MYOD and other myogenic factors (reviewed in [189]). In vertebrates, there are four MEF2 paralogs: MEF2A-D, each encoded by a distinct gene. MEF2 proteins share a N-terminal DNA-binding and dimerization domain, while the transcripts are highly diversified and undergo extensive alternative splicing within their C-terminal transactivation domain, which produce multiple isoforms. MEF2 splice variants differently participate in early commitment to muscle differentiation and maintenance of the differentiated state in vertebrates [190,191,192,193,194]. In addition to alternative splicing, the activity of MEF2 factors is finely modulated by various means that directly involve their transcripts. It has indeed been shown that translation of *Mef2ca* in zebrafish is negatively regulated by interaction of the transcript with eukaryotic initiation factor 4E binding proteins (eIF4EBPs) in response to inactivity [195]. Further, the murine *Mef2a* transcript is subjected to a translational control mechanism that is mediated by the *Mef2a* 3′UTR which is relieved during muscle cell differentiation [196]. The molecular details of these regulatory processes are still waiting to be clarified, we can hypothesize that dynamic chemical modifications of *Mef2* transcripts might play a role, possibly modulating the interaction with trans-acting RNA binding factors in response to external stimuli. The hypothesis that the *Mef2* transcripts undergo a modulation of their chemical modification profile during muscle differentiation is further strengthened by literature data which demonstrate METTL3 stabilizes *Mef2c* RNA and increases its translation in bovine and quail muscle cells [163,197]. Accordingly, multiple m6A and A-to-I sites have been retrieved within the *Mef2* transcripts (Table 3). Interestingly, only the *Mef2a* transcript shows two pseudouridine (*ψ*) modifications. It is well known that *ψ* is the most abundant modified nucleoside in non-coding RNAs, stabilizing the tRNA and rRNA structure and enhancing their function [101]. In addition, *ψ* regulates the splicing process by modifying specific snRNAs, while its role in mRNA remains essentially unknown [198]. Mutations in genes encoding for pseudouridine synthases have been identified in patients with neurodevelopmental disorders [199,200], thus reinforcing the possible role of *ψ* as a regulator also in the expression of neuromuscular genes, such as *Mef2a*. Recently, *ψ* modifications have been identified in nascent pre-mRNA at sites associated with alternative splicing [104]; therefore, it would be useful to investigate whether a correlation between *ψ* and *Mef2a* splicing exists. Moreover, since *ψ* content in 3′-UTR mRNA is regulated in response to environmental signals [101], flexible adaptation to continuous neighborhood environmental factors in pathological muscle may be induced through *ψ* mRNA modifications. Dysregulated expression and splicing of *Mef2a*, *Mef2c* and *Mef2d* genes occur in several neuromuscular disorders, including Becker syndrome and myotonic dystrophy (DM), which is characterized by the expression of *MEF2* embryonic isoforms [201,202,203]. Therefore, clarifying the impact of RNA modifications on MEF2 function will potentially open new therapeutic options for these pathologies.

Another gene that controls muscle cell fate is nuclear transcription factor-Y alpha (*Nfya*), which encodes for the NF-YA DNA binding subunit of the transcription factor NF-Y (Nuclear Transcription Factor Y). Expression levels and alternative splicing of NF-YA are crucial in muscle cell proliferation and differentiation, and muscle stem cell-specific knockout studies highlighted that NF-YA expression is fundamental to preserve the pool of muscle stem cells and ensures muscle regeneration upon injury [204,205,206]. Although NF-YA mutations have not been observed in muscular diseases, we cannot exclude that alterations in its expression or splicing may participate in muscle pathogenetic mechanisms, as demonstrated in cancer disease [207,208]. Indeed, two alternative splice isoforms are generated from the *Nfya* gene (NF-YAs and NF-YAl) and are not functionally equivalent in various types of cells, myoblasts included [204]. In mouse embryonic myoblasts, the expression of both NF-YA variants is high and drops in post-natal muscles, with only NF-YAl being expressed at low levels [205]. Gain of function studies highlighted that NF-YAs enhances cell proliferation, in opposition to NF-YAl that improves cell differentiation [204]; therefore, it would be key to investigating whether epitranscriptomics mechanisms participate to *Nfya* splicing or can be exploited to restore altered events. Finally, an interesting crosstalk between m6A and NFY came to light from studies on myeloid leukaemia [209]. The interrogation of sequences under METTL3 peaks for enriched motifs identified the NF-Y binding site as the top hit, suggesting a cooperative binding between METTL3 and NF-Y on promoter-bound METTL3 to maintain m6A-dependent translation control.

SMAD4 has been included in our analysis because of its role in muscle stem cells activity: it is a downstream cofactor for canonical TGFβ superfamily signaling, and *Smad4*-specific deletion in adult mouse stem cells triggers terminal myogenic commitment associated with impaired proliferative potential. Consistently, adult skeletal muscle regeneration is evidently compromised following *Smad4* abrogation [210]. Through the interaction with AR, SMAD4 chromatin binding orchestrates a muscle hypertrophy transcriptional program that is altered in the spinal and bulbar muscular atrophy (SBMA) mouse model [211]. Understanding the possible diverse set of modifications or rewiring the modifications in *Smad4* transcript may be crucial in the fine-tune of AR-SMAD4 functional complex, which has been proposed as a promising target for SBMA and other conditions associated with muscle loss.

Overall, deep analysis of RNA modifications of muscle-related gene transcripts in pathological contexts may provide new mechanisms for the basis of altered expression and splicing events. The most challenging question before the potential application of therapies targeting epitranscription in muscular diseases is to monitor the dynamic changes in RNA modifications, such as m6A, under physiological and pathological conditions.

## Figures and Tables

**Figure 1 ijms-24-15161-f001:**
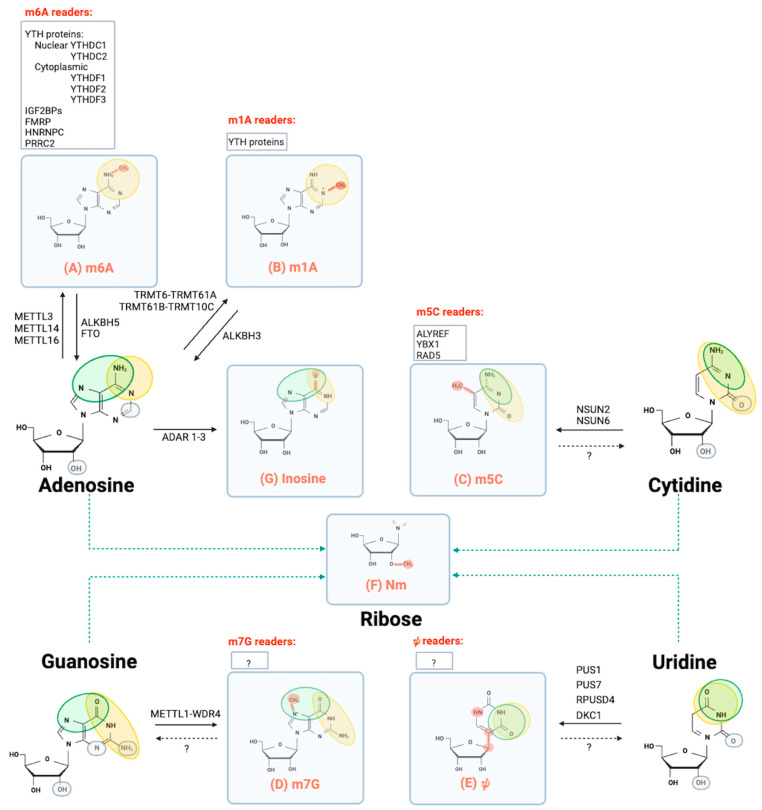
Chemical structure of mRNA modifications reported in eukaryotic. The four standard ribonucleotides are reported with ovals indicating the Watson–Crick (green), Hoogsteen (orange) and sugar (grey) edges of the ribonucleotides. In the squares are indicated the chemical modifications described in the text: N⁶-methyladenosine (m6A), N^1^-methyladenosine (m1A), inosine, 5-methylcytidine (m5C), 7-methylguanosine (m7G), 2′O-methylation (Nm), pseudouridine (*ψ*). A summary of the known writers and erasers is also reported, known readers are listed in the grey squares. The question marks and dotted arrows indicate the reactions catalyzed by enzymes that have not yet been identified. Question marks in grey squares denote the lack of knowledge about readers of the chemical modification represented below the squares.

**Figure 2 ijms-24-15161-f002:**
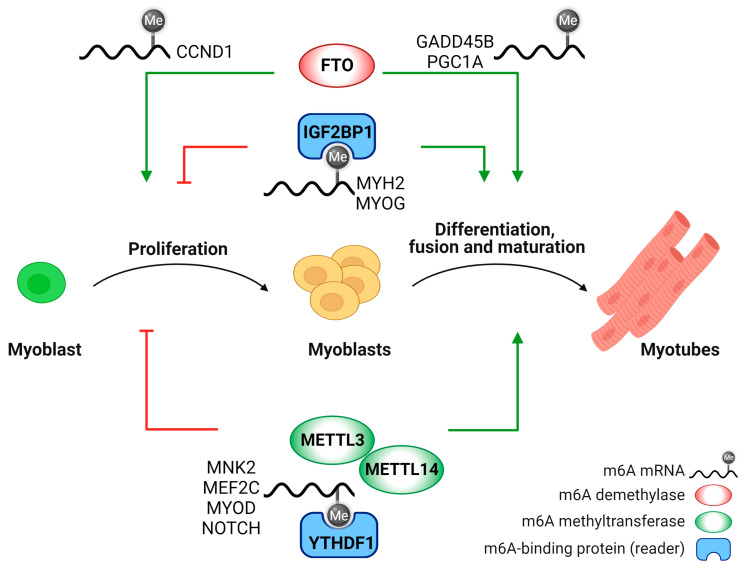
m6A RNA modification during myoblasts proliferation and differentiation. Writers, erasers and readers of m6A RNA modification post-transcriptionally regulate genes involved in myogenesis, thus affecting the transition from myoblast proliferation to differentiation. For possible control and genes involved, see Section 2.2.

**Table 1 ijms-24-15161-t001:** Stoichiometry, consensus sequence and mRNA preferred regions of modified ribonucleosides. N stands for any base, D = A, G, or U; H = A, C or U; R = G or A; Y = C or U. Underlined nucleotides in the consensus sequences are chemically modified.

Modified Ribonucleoside	Stoichiometry	Target Sequence	mRNA Preferred Regions
m6A	0.2–0.6% m6A/A	DRACH	5′ UTR, near the stop codon, 3′ UTR
m1A	0.01–0.05% m1A/A	GC-rich and GA-rich sequence, GUUCNANNC	All segments of the transcripts, enriched in: first splice site, GC-rich highly structured regions in the 5′ UTRs, translation start site
m5C	0.03–0.1% m5C/C	CTCCA	5′/3′ UTRs, next to argonaute-binding regions
m7G	0.02–0.05% m7G/G	GA- or GG-enriched sequence motifs AG-rich sequence	5′UTR near the start codon 3′UTR
*ψ*	0.2–0.6% *ψ*/U	AU (PUS1-specific) GUUCNANYCY (PUS4) UGUAG PUS7)	Coding sequence and 3′UTR.
Nm	0.01% Nm/N	NmAGAUC followed by a 5 nt-long AG-rich stretch	CDS, near splice sites, 5′/3′ UTR, introns, alternatively spliced regions
A to I	Not available	dsRNA, preference for U in the −1 position	Introns (Alu sequences) and UTRs; coding sequence

**Table 2 ijms-24-15161-t002:** List of RNA modifications identified from high-throughput epitranscriptome sequencing data collected in the RMBase v3.0 database [17] (https://rna.sysu.edu.cn/rmbase3/, accessed on 30 May 2023). Genes related to MD were obtained from the 2023 GeneTable of Neuromuscular Disorders [19] (http://www.musclegenetable.fr, accessed on 30 May 2023). The numbers represent the number of RNA modification sites of a specific modification type on the gene.

Gene	Gene ID	m6A	m1A	m5C	m7G	PseudoU	2′-O-Me	RNA-Editing, A-I Sites	Total
*ACTA1*	ENSG00000143632.14	1	0	0	0	0	0	2	3
*ANO5*	ENSG00000171714.11	23	0	0	0	0	0	3	26
*B3GALNT2*	ENSG00000162885.13	13	0	1	0	0	1	8	23
*B4GAT1*	ENSG00000174684.7	27	1	1	0	0	0	0	29
*BVES*	ENSG00000112276.14	11	0	0	0	0	0	1	12
*CACNA1S*	ENSG00000081248.11	6	0	2	0	0	0	0	8
*CAPN3*	ENSG00000092529.24	17	0	0	0	0	0	2	19
*CAV3*	ENSG00000182533.6	7	0	0	0	0	0	0	7
*CAVIN1*	ENSG00000177469.13	45	0	1	0	0	0	0	46
*CHKB*	ENSG00000100288.19	25	0	0	0	0	0	1	26
*COL12A1*	ENSG00000111799.21	203	0	0	0	0	0	0	203
*COL6A1*	ENSG00000142156.14	93	0	0	0	0	0	0	93
*COL6A2*	ENSG00000142173.15	88	0	0	0	0	0	3	91
*COL6A3*	ENSG00000163359.15	144	0	1	0	0	0	5	150
*DAG1*	ENSG00000173402.11	138	0	2	0	0	1	6	147
*DES*	ENSG00000175084.11	35	0	1	0	0	0	0	36
*DMD*	ENSG00000198947.15	116	0	2	0	0	0	19	137
*DNAJB6*	ENSG00000105993.15	86	0	14	0	0	0	8	108
*DNM2*	ENSG00000079805.16	118	0	13	0	1	0	29	161
*DPM1*	ENSG00000000419.12	25	0	0	0	0	1	16	42
*DPM2*	ENSG00000136908.17	54	0	0	0	0	0	0	54
*DPM3*	ENSG00000179085.7	6	0	1	0	0	0	0	7
*DYSF*	ENSG00000135636.14	33	0	4	0	0	0	1	38
*EMD*	ENSG00000102119.10	22	0	10	0	0	0	0	32
*FHL1*	ENSG00000022267.17	76	0	0	0	4	0	2	82
*FKRP*	ENSG00000181027.10	64	0	0	0	0	0	12	76
*FKTN*	ENSG00000106692.14	49	0	0	0	1	0	4	54
*GAA*	ENSG00000171298.13	69	0	6	0	2	0	0	77
*GGPS1*	ENSG00000152904.11	83	0	5	0	0	0	1	89
*GMPPB*	ENSG00000173540.12	76	0	0	0	0	0	1	77
*GOLGA2*	ENSG00000167110.17	81	0	3	0	0	0	10	94
*GOSR2*	ENSG00000108433.16	139	0	0	0	0	0	3	142
*HNRNPDL*	ENSG00000152795.17	94	0	3	0	0	1	0	98
*INPP5K*	ENSG00000132376.20	62	0	1	0	0	0	1	64
*ITGA7*	ENSG00000135424.16	39	0	0	0	0	0	0	39
*JAG2*	ENSG00000184916.9	48	0	4	1	0	0	0	53
*LAMA2*	ENSG00000196569.12	21	0	0	0	0	1	3	25
*LARGE1*	ENSG00000133424.20	62	0	0	0	0	1	63	126
*LIMS2*	ENSG00000072163.19	27	0	0	0	0	0	2	29
*LMNA*	ENSG00000160789.20	72	0	5	0	0	1	22	100
*LRIF1*	ENSG00000121931.16	72	0	0	0	0	0	0	72
*MPDU1*	ENSG00000129255.16	36	0	2	0	0	0	0	38
*MSTO1*	ENSG00000125459.15	33	0	1	0	1	0	0	35
*MYOT*	ENSG00000120729.9	7	0	0	0	0	0	0	7
*PLEC*	ENSG00000178209.15	167	0	26	1	2	0	21	217
*POGLUT1*	ENSG00000163389.12	37	0	1	0	1	0	1	40
*POMGNT1*	ENSG00000085998.14	59	0	1	0	0	0	0	60
*POMGNT2*	ENSG00000144647.6	49	0	1	0	0	0	1	51
*POMK*	ENSG00000185900.9	24	0	0	0	0	0	0	24
*POMT1*	ENSG00000130714.16	53	0	4	0	0	0	16	73
*POMT2*	ENSG00000009830.11	66	0	2	0	0	0	5	73
*POPDC3*	ENSG00000132429.10	25	0	0	0	0	0	0	25
*PYROXD1*	ENSG00000121350.16	48	0	1	0	2	0	0	51
*RXYLT1*	ENSG00000118600.11	23	0	2	0	0	0	1	26
*RYR1*	ENSG00000196218.12	32	0	3	0	0	1	1	37
*SELENON*	ENSG00000162430.17	59	0	4	0	0	0	15	78
*SGCA*	ENSG00000108823.16	3	0	0	0	0	0	0	3
*SGCB*	ENSG00000163069.12	61	0	17	0	0	0	2	80
*SGCG*	ENSG00000102683.7	6	0	0	0	0	0	0	6
*SMCHD1*	ENSG00000101596.15	153	0	2	0	0	0	16	171
*SYNE1*	ENSG00000131018.23	339	0	4	0	0	2	50	399
*SYNE2*	ENSG00000054654.16	482	0	2	0	0	0	22	506
*TCAP*	ENSG00000173991.5	11	0	4	0	0	0	0	15
*TMEM43*	ENSG00000170876.7	89	0	2	0	0	0	0	91
*TNPO3*	ENSG00000064419.13	94	0	1	0	0	0	8	103
*TOR1AIP1*	ENSG00000143337.18	90	0	5	0	0	0	3	98
*TRAPPC11*	ENSG00000168538.16	67	0	0	0	0	0	3	70
*TRIM32*	ENSG00000119401.10	38	0	0	0	1	0	0	39
*TRIP4*	ENSG00000103671.9	23	0	0	0	0	0	11	34
*TTN*	ENSG00000155657.26	249	0	5	0	0	1	0	255
*VCP*	ENSG00000165280.16	84	0	10	0	0	6	0	100

**Table 3 ijms-24-15161-t003:** Lists of genes encoding for myogenic regulatory factors and related RNA modifications from the RMBase v3.0 database [17] (https://rna.sysu.edu.cn/rmbase3/, accessed on 30 May 2023)). The numbers represent the number of RNA modification sites of a specific modification type on the gene.

Gene	Gene ID	m6A	m1A	m5C	m7G	PseudoU	2′-O-Me	RNA-Editing, A-I Sites	Total
*EZH2*	ENSG00000106462.10	52	0	0	0	0	1	4	57
*HDAC4*	ENSG00000068024.16	80	0	17	0	0	0	12	109
*MEF2A*	ENSG00000068305.17	128	0	1	0	2	0	49	180
*MEF2C*	ENSG00000081189.15	41	0	0	0	0	0	3	44
*MEF2D*	ENSG00000116604.18	38	0	2	0	0	0	8	48
*MYF5*	ENSG00000111049.4	0	0	0	0	0	0	0	0
*MYH1*	ENSG00000109061.10	0	0	0	0	0	0	1	1
*MYH2*	ENSG00000125414.19	2	0	1	0	0	0	0	3
*MYOD1*	ENSG00000129152.4	0	0	0	0	0	0	0	0
*NFYA*	ENSG00000001167.14	79	0	6	1	0	0	1	87
*PAX3*	ENSG00000135903.19	59	0	0	0	0	0	0	59
*PAX7*	ENSG00000009709.12	37	0	0	0	0	0	0	37
*SIRT1*	ENSG00000096717.12	76	0	0	0	0	0	27	103
*SMAD4*	ENSG00000141646.13	104	0	4	0	0	0	5	113

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
