# Peer review of "Epitranscriptomics as a New Layer of Regulation of Gene Expression in Skeletal Muscle: Known Functions and Future Perspectives"

_ijms, 2023, doi:10.3390/ijms242015161_

Round 1

Reviewer 1 Report

Dear,

 A very clear overview of the current knowledge about functional roles of epitranscriptomic marks in the
skeletal muscle system. The introduction about epitransciptomics is very clear and informative.   Some typos were found: line 76, "epitransciptomics"                                                     line 215: please remove the extra space                                                     line 222: please check "cytines"                                                     line 259: please remove "in"                                                     line 301: please check "rev in"                                                     line 316: please close the brackets                                                     line 342: please put latin name in italic                                                     line 365: please put th etilt ein bold and remove a space                                                     line 366: Watson-Crick                                                     line 370: please remove the bracket                                                     line 371: please put the title in bold                                                     line 377: please put the gene in italic                                                     line 381: idem                                                     line 459: please provide the year                                                     line 576: please foresee a space before the bracket                                                     line 625: idem                                                     line 631: please write full out "review"     Please write all abbreviations full out in the text and provide a list of abbreviations  

Figure 2: please provide explanation under the title of the figure.Please enlarge the figure.

May I please ask for a list of abbreviations?

Best,

Reviewer

Just some typos to check.

Author Response

We really appreciate the reviewer's interest in our work and the positive comment. We are sorry for any mistakes and typos. We have fixed them. As suggested by the reviewer we have modified the text and added the list of abbreviatons. We thank the reviewer for his/her efforts to give us suggestions to ameliorate our manuscript. 

Reviewer 2 Report

This is a well written informative review that summarises current knowledge of RNA modifications, focusing in particular on skeletal muscle and possible therapeutic opportunities. I believe this article is of sufficient quality for publication with only minor editorial changes required.

Minor comments:

Please clarify the sentence starting in line 36:

“The variety of chemical modifications observed in RNAs is much wider than epigenetic modifications detected in genomic DNA, possibly because they confer a high grade of versatility to the RNA molecules that can perform a large number of regulatory and catalytic functions through the assumption of specific folds.”

This sentence suggests that there are more chemical modifications found in RNA than in DNA and this is because the chemical modifications in RNA confer a large number of regulatory and catalytic functions. Is this what the authors mean? Could the authors clarify the causality in this sentence and perhaps provide a reference to support this claim?

Please provide a reference for this statement: “For example, base modifications were first identified in abundant RNA species like tRNAs and rRNAs at the end of the 1960s, and their possible role in RNA processing and size reduction has been inferred.”

Line 234: Please correct the gramma in this sentence.

Line 242: embryonic stem cell differentiation

Line 366: Watson-Crick instead of Watson Creek

Figure 1: This figure is very informative, but the writing is very small and hard to read. Perhaps the size of images and writing can be increased.

Well written article.

Author Response

We sincerely thank the Reviewer for the positive comment.

 In accordance to the suggestion of the reviewer we made an effort to clarify the sentence starting from line 36 adding also a reference describing the lower number of chemical modifications that have been described so far for genomic DNA (Raiber et al 2017). We also added some references describing the first chemical modifications in tRNA and rRNA in 1960s (Iwanami et al 1968, RajBhandary et al 1966, Dunn et al 1961).  We also correted the gramma in sentence of line 234 according to the Reviewer’s suggestion, We apologize for typos that we have fixed. We changed figure 1 size and increased the characters, accordingly.

We again thank the reviewer for his/her useful suggestions.

Reviewer 3 Report

This is a  very nice and well structured review in an important and fast-moving field. The authors first review in depth the RNA modifications in general, and then focus on skeletal muscle. The given web-site links to data bases like ENCOR make this review a very valuable resource to look then up genes of interest.  I have only one  ver minor critique/suggestion. At this stage it still seems to be unclear for many modification events if they  control/contribute more to homeostasis, or differentiation, or fusion, per perhaps intercell communications (fibroblasts/myoblasts, etc).  I therefore recommend to down-tune  the direct control conclusion in the legend to Figure 2 (and perhaps sometimes also in the text):

Legend to Figure 2.  "RNA modification controls myoblasts proliferation and differentiation  through post-499 transcriptional regulation of genes involved in myogenesis."

Suggestion  to revise to a more convervative and descriptive  statement like "m6A RNA modification DURING myoblasts proliferation and differentiation.

For possible control and genes involved, see below section 2.3" 

Author Response

We really thank the reviewer for his/her positive comments on our manuscript.

We sincerely thank the Reviewer for the positive comment to our Review. We agree with the suggestions, accordingly, we toned down some conclusions throughout the text and we have changed figure legend. We again thank the reviewer for the precious advises.